# Laherradurin Inhibits Colorectal Cancer Cell Growth by Induction of Mitochondrial Dysfunction and Autophagy Induction

**DOI:** 10.3390/cells13191649

**Published:** 2024-10-03

**Authors:** Izamary Delgado-Waldo, Svetlana Dokudovskaya, Yahir A. Loissell-Baltazar, Eduardo Pérez-Arteaga, Jossimar Coronel-Hernández, Mariano Martínez-Vázquez, Eloy Andrés Pérez-Yépez, Alejandro Lopez-Saavedra, Nadia Jacobo-Herrera, Carlos Pérez Plasencia

**Affiliations:** 1Unidad de Bioquímica, Instituto Nacional de Ciencias Médicas y Nutrición Salvador Zubiran, Av. Vasco de Quiroga 15, Col. Belisario Domínguez Sección XVI, Tlalpan, Ciudad de México 14080, Mexico; izz.waldo11@gmail.com (I.D.-W.); epa.biol@outlook.es (E.P.-A.); 2Posgrado en Ciencias Biológicas, Universidad Nacional Autónoma de México, Copilco Universidad, Coyoacán, Ciudad de México 04510, Mexico; 3CNRS UMR9018, Institut Gustave Roussy, Université Paris-Saclay, 94805 Villejuif, France; svetlana.dokudovskaya@gustaveroussy.fr (S.D.); y.loissell@gmail.com (Y.A.L.-B.); 4Laboratorio de Genómica, Instituto Nacional de Cancerología, Instituto Nacional Nacional de Cancerología, Av. San Fernando 22, Belisario Domínguez Secc 16, Tlalpan, Ciudad de México 14080, Mexico; jossithunders@gmail.com (J.C.-H.); eperezy2306@gmail.com (E.A.P.-Y.); 5Instituto de Química, Universidad Nacional Autónoma de México, C. Exterior, C. Universitaria, Coyoacán, Ciudad de México 04510, Mexico; marvaz@unam.mx; 6Advanced Microscopy Applications Unit (ADMIRA), Instituto Nacional de Cancerología, San Fernando 22. Col. Sección XVI, Tlalpan, Ciudad de México 14080, Mexico; alex@admiramicro.com; 7Escuela de Medicina y Ciencias de la Salud, Tecnológico de Monterrey Ciudad de Mexico, C. Puente #222, Coapa, Arboledas del Sur, Tlalpan, Ciudad de Mexico 14380, Mexico; 8Laboratorio de Genómica Funcional, Unidad de Biomedicina, Facultad de Estudios Superiores Iztacala, UNAM, Tlalnepantla Estado de México 54090, Mexico

**Keywords:** laherradurin, acetogenins, colon cancer, mitochondria, autophagy, apoptosis, glycolysis

## Abstract

LAH, an acetogenin from the Annonaceae family, has demonstrated antitumor activity in several cancer cell lines and in vivo models, where it reduced the tumor size and induced programmed cell death. We focused on the effects of LAH on mitochondrial dynamics, mTOR signaling, autophagy, and apoptosis in colorectal cancer (CRC) cells to explore its anticancer potential. Methods: CRC cells were treated with LAH, and its effects on mitochondrial respiration and glycolysis were measured using Seahorse XF technology. The changes in mitochondrial dynamics were observed through fluorescent imaging, while Western blot analysis was used to examine key autophagy and apoptosis markers. Results: LAH significantly inhibited mitochondrial complex I activity, inducing ATP depletion and a compensatory increase in glycolysis. This disruption caused mitochondrial fragmentation, a trigger for autophagy, as shown by increased LC3-II expression and mTOR suppression. Apoptosis was also confirmed through the cleavage of caspase-3, contributing to reduced cancer cell viability. Conclusions: LAH’s anticancer effects in CRC cells are driven by its disruption of mitochondrial function, triggering both autophagy and apoptosis. These findings highlight its potential as a therapeutic compound for further exploration in cancer treatment.

## 1. Introduction

CRC is the third most prevalent malignant neoplasm worldwide, affecting individuals of both sexes. The treatment options for CRC vary depending on the tumor’s location and the stage at diagnosis, and may include surgery, chemotherapy, and radiation therapy [1]. One of the main issues with chemotherapeutic agents is their toxicity and the development of drug resistance. Consequently, there is a continuous effort to discover new compounds that offer enhanced efficacy and reduced toxicity compared to conventional treatments.

In cancer drug discovery, herbs are used frequently as a source of new molecules [2]. Plant metabolites, renowned for their structural diversity and bioactivity, have garnered significant attention for their anticancer properties [3]. Plant secondary metabolites, including phenanthrenes, phenols, flavonoids, bibenzyls, alkaloids, phenols, anthocyanins, and acetogenins, are bioactive molecules that may function as antioxidant, anti-inflammatory, antimicrobial, antiparasitic, anticoagulant, antidiabetic, and anticancer agents [4]. In this context, the first plant-derived antitumor agent approved by the FDA, vinicristine, is used to treat lymphoblastic leukemia [5], while other plant-derived drugs including but not limited to paclitaxel are applied for the treatment of breast, ovarian, and lung cancer [6], and omacetaxine mepesuccinate (homoharringtonine) is used for chronic myeloid leukemia [7], among others.

The annonaceous acetogenins (ACGs), obtained from the Annonaceae family, are a class of specialized metabolites with potential anticancer properties [8]. Notably, LAH (Figure 1A), an ACG extracted from the seeds of Annonaspecies, demonstrated significant cell growth inhibition in CRC models [9]. Previously our research group examined the antineoplasic effect of LAH in a colitis-associated tumor model in mice. We found that LAH reduced the tumor volume and number, suppressed cell migration in vitro, and induced cell death via apoptosis [10].

The principal molecular mechanism of ACGs involves the impairment of mitochondrial NADH-ubiquinone oxidoreductase (complex I), leading to adenosine triphosphate (ATP) depletion and cell cycle arrest [11]. Mitochondria are important for energy production, apoptosis regulation, metabolic reprogramming, and reactive oxygen species (ROS) generation. The dysregulation of mitochondrial dynamics, characterized by alterations in morphology, function, and signaling pathways, has been associated with cancer development [9]. In addition, mitochondrial dysfunction is related to cancer cell survival, metastasis, and resistance to therapy, making them attractive targets for anticancer interventions [12]. The modulation of mitochondrial dynamics through plant-derived secondary metabolites represents a novel approach to induce apoptosis [13].

Recently, ACGs were shown to [13] induce autophagy in cancer cells [9,10,14,15,16,17,18,19]. Autophagy, a major catabolic process in the cells, is under the control of the PI3K/AKT/mTOR signaling pathway, a crucial regulator of cell proliferation and survival. The inhibition of mTOR complex 1 (mTORC1), leading to autophagy induction, has emerged as a novel anti-cancer strategy [20].

Our main goal was to evaluate the antitumor potential of LAH. To do so, we confirmed that it inhibits the electron transport chain complex. We found an increase in glycolysis and an oxidative phosphorylation (OXPHOS) activity reduction due to the inhibition of mitochondrial complex I at the metabolic level. We hypothesized that LAH induces damage at the mitochondrial level, which we demonstrated with a diminished mitochondrial membrane potential in addition to an elevation in mitochondrial mass due to mitochondrial fission and mitophagy induction. On the other hand, LAH, while affecting the glycolytic balance and OXPHOS, had a specific effect on autophagic flux in tumor cells in contrast to non-tumoral cells exposed to LAH. Finally, we demonstrated that these metabolic changes induce DNA damage leading to cell death via apoptosis

## 2. Materials and Methods

### 2.1. Reagents

LAH was extracted from the seeds of *Annona diversifolia* Saff (Annonaceae), collected in Copoya, Chiapas, México, and identified at the Eizi Matuda Herbarium of the Universidad de Ciencias y Artes de Chiapas, México. Additional controls included Rapamycin (RAP) at 1 µM (TO-R001, Euromedex, Souffelweyersheim, France), Hydroxychloroquine sulfate (HCQ) at 50 µM (HY-B1370, MedChemExpress, Monmouth Junction, NJ, USA), TORIN-1 at 1 µM (HY-13003, MedChemExpress, Monmouth Junction, NJ, USA), Oligomycin A at 10 µM (HY-16589, MedChemExpress, Monmouth Junction, NJ, USA), CCCP at 100 µM (C2759, Sigma-Aldrich, St. Louis, MO, USA), Antimycin A at 2.5 μM (A8674, Sigma-Aldrich, St. Louis, MO, USA), glucose at 20 mM (103577-100, Agilent Technologies, Santa Clara, CA, USA), and 2-DG at 20 nM (103344-100, Agilent Technologies, Santa Clara, CA, USA).

### 2.2. Cell Culture

The cell lines, HCT116 and SW620, and non-tumor cell line, CRL-1790, were provided by the National Cancer Institute in Mexico City, México. The cells were maintained in 100 × 15 mm culture dishes in DMEM/F-12 GIBCO^®^ medium (Thermo Fisher Scientific, Waltham, MA, USA) supplemented with 10% fetal bovine serum (FBS, GIBCO^®^, Thermo Fisher Scientific, Waltham, MA, USA) at 37 °C with 5% CO₂. At 80% confluence, the cells were exposed to the treatments.

### 2.3. Cytotoxicity Assay

The IC_50_ value for LAH was determined via a Sulforhodamine B (SRB) colorimetric assay following the protocol established in 2006 [21]. Briefly, cells were seeded in 96-well plates at a density of 1 × 10^4^ cells/well, allowed to adhere overnight and subsequently treated with different doses of LAH for 24 h. The cells were fixed with cold 10% trichloroacetic acid at 4 °C for 1 h and washed four times with water, followed by staining with 100 μL of 0.5% SRB (S9012, Sigma-Aldrich, St. Louis, MO, USA) in 1% acetic acid for 30 min at room temperature. The excess of the stain was removed via washing four times with 1% acetic acid. The cells were resuspended in 200 mL of 10 mM Tris base at pH 10. The optical density at 510 nm was determined by using an Epoch microplate spectrophotometer (BioTek, Winooski, VT, USA). Dose–response curves and cellular viability values were generated via GraphPad Prism software 8.0.2. All the experiments were performed in triplicate.

### 2.4. Western Blotting

Cell pellets were resuspended in NETN lysis buffer, and protein concentrations were determined. Samples were loaded onto 4–12% Tris-Glycine gels and transferred to PVDF membranes (IPVH00010, Millipore, Burlington, MA, USA). After blocking in 5% milk in TBS-T, membranes were incubated with primary antibody overnight at 4 °C, and membrane was washed 3 times with TBS-T followed by incubation with an appropriate secondary antibody for one hour at room temperature. Protein bands were visualized using an enhanced chemiluminescent substrate (WBKLS0500, Millipore, Burlington, MA, USA). The following antibodies were used: total OXPHOS human WB antibody cocktail (ab110411, Abcam, Cambridge, UK); anti-β-actin (C4) (sc-47778, Santa Cruz Biotechnology, Dallas, TX, USA); anti-caspase-8 (AM46, Millipore, Burlington, MA, USA); anti-mitofusin 1 (14739S), anti-phospho-DRP1 (Ser637) (6319S), anti-DRP1 (8570), anti-phospho-4E-BP1 (Thr37/46) (2855S), anti-4E-BP1 (9644S), anti-ULK1 (8054S), anti-phospho-ULK1 (Ser757) (14202), anti-P62 (5114S), anti-LC3B (2775), anti-PARP1 (9542), anti-cleaved PARP (9541), anti-AIF (5318), anti-caspase-3 (9662), anti-GAPDH (51332S), anti-mouse IgG, HRP-linked (7076), anti-rabbit IgG, HRP-linked (7074) were from Cell Signaling Technology, Danvers, MA, USA.

### 2.5. ATP Measurement

ATP concentrations were measured in triplicate using the ATPlite 1step Luminescence Assay System kit (6016736, PerkinElmer, Waltham, MA, USA) following the manufacturer’s protocol. Total luminescence was measured using a 96-well microplate reader. ATP levels were expressed as a percentage of the untreated control.

### 2.6. Measurement of Mitochondrial Respiration

Mitochondrial respiration was assessed using the Seahorse XF96 extracellular flux analyzer (Agilent Technologies, Santa Clara, CA, USA). The cells were seeded in Seahorse XF96 cell culture microplates and allowed to adhere for 24 h. After exposure to the treatments, the cells were washed and incubated with XF RPMI medium. The compounds from the Seahorse XF Cell Mito Stress Test Kit (103015-100, Agilent Technologies, Santa Clara, CA, USA), and Seahorse XF Glycolytic Rate Assay Kit (103344-100, Agilent Technologies, Santa Clara, CA, USA) were loaded onto the microplate, and measurements were performed using Seahorse Wave software 2.6.3.5. The data were normalized based on the fluorescence intensity with Hoechst staining and the quantification of the total protein by using a bicinchoninic acid (BCA) assay (23227, Pierce BCA Protein Assay Kit, Thermo Fisher Scientific, Waltham, MA, USA).

### 2.7. Mitochondrial Membrane Potential Assessment

The mitochondrial transmembrane potential was assessed using Tetramethylrhodamine methyl ester (TMRM) (T5428, Sigma-Aldrich, St. Louis, MO, USA) fluorescence. CRC cells were seeded in 6-well plates and allowed to adhere for 24 h. Following the treatment with LAH, the cells were washed and incubated in DMEM containing 100 nM TMRM for 10 min at 37 °C in the dark. For flow cytometry analysis, TMRM fluorescence was excited at 549 nm and measured using a BD Accuri™ C6 Plus flow cytometer (BD Biosciences, Franklin Lakes, NJ, USA). A total of 1 × 10^4^ cells were analyzed per condition. The relative TMRM intensity was calculated using FlowJo X software 10.8.1. Baseline TMRM levels were independently determined for each cell line, and these levels were used for normalization. To validate the mitochondrial membrane potential, the cells were treated with 10 μM Oligomycin A as a positive control and 10 µM CCCP as a negative control.

### 2.8. Analysis of Mitochondrial Morphology

The cells were seeded on poly-D-lysine-treated coverslips in 6-well plates and allowed to adhere for 24 h. After treatment with LAH, the cells were incubated with RPMI containing MitoTracker Orange CMTMRos (M7510, Thermo Fisher Scientific, Waltham, MA, USA) for 45 min at 37 °C in the dark. Following incubation, the cells were fixed with 4% paraformaldehyde (PFA), washed, and mounted on glass slides with Fluoroshield Mounting Media containing DAPI (F6057, Sigma-Aldrich, St. Louis, MO, USA). Images were captured at an excitation wavelength of 554 nm using a Confocal Leica SPE DM4000B microscope (Leica Microsystems, Wetzlar, Germany) with a 63× oil-immersion objective.

### 2.9. Mitochondrial Mass Measurement 

The CRC cells were seeded in 6-well plates and treated with LAH. Following the treatment, the cells were incubated with 10% FBS RPMI medium containing 100 nM 10-N-nonyl acridine orange (NaO) (A1301, Invitrogen, Waltham, MA, USA) for 10 min at 37 °C in the dark. The fluorescence intensity was measured using a BD Accuri™ C6 Plus flow cytometer (BD Biosciences, Franklin Lakes, NJ, USA), with 1 × 10^4^ cells analyzed per condition. The relative NaO intensity was calculated using FlowJo X software 10.8.1, baseline NaO levels were independently determined for each cell line, and these levels were used for normalization. Oligomycin A and CCCP were used as positive and negative controls for the mitochondrial membrane potential, respectively.

### 2.10. Immunofluorescence Staining

To localize autophagosomes, the cells were transfected with the pBABE-puro mCherry-EGFP-LC3B plasmid (22418, Addgene, Cambridge, MA, USA). The cells were cultured in 24-well plates on coverslips coated with 0.1 mg/mL poly-D-lysine (A-003-E, Sigma-Aldrich, St. Louis, MO, USA) at a density of 4 × 10^4^ cells per well and allowed to adhere overnight at 37 °C. Subsequently, the cells were transfected with 1 µg of plasmid DNA using Lipofectamine 2000 (11668019, Thermo Fisher Scientific, Waltham, MA, USA) and subjected to drug treatment with LAH for 24 h. The cells were then fixed with 4% PFA for 30 min and permeabilized with 0.1% Triton X-100 in phosphate-buffered saline (PBS-T: 137 mM NaCl, 2.7 mM KCl, 10 mM Na_2_HPO_4_, 2 mM KH_2_PO_4_, pH 7.4) for 5 min. Coverslips were blocked with 1% bovine serum albumin (BSA) in PBS-T for 1 h. For immunostaining, the coverslips were incubated overnight with the primary antibody against alpha-tubulin (DM1A, 1:100, Cell Signaling Technology, Danvers, MA, USA). After washing with PBS-T, the coverslips were incubated with Alexa Fluor 568-conjugated mouse secondary antibody (ab175473, 1:100, Abcam, Cambridge, UK) in 1% BSA in PBS-T for 1 h at room temperature. After three washes with PBS-T, the coverslips were incubated for 5 min with PBS-T containing 10 µM DAPI. Finally, the coverslips were mounted with Vectashield Vibrance (H-1700, Vector Laboratories, Burlingame, CA, USA) on slides and sealed with nail polish. Digital images were captured using a Zeiss LSM 880 inverted microscope (Oberkochen, Germany) with a 63× oil-immersion objective at the appropriate wavelength (excitation at 587 nm for mCherry, excitation at 568 nm for tubulin, and excitation at 405 nm for DAPI) and analyzed using FIJI software 1.53. In order to detect γ-H2AX, all the manipulations were made as above (without plasmid transfection and using phalloidin to stain the cytoskeleton. Anti-γ-H2AX (D17A3) (Cell Signaling Technology, Danvers, MA, USA) (1:100) was used as a primary antibody, and goat-anti rabbit Cy2-conjugated was used as a secondary antibody (ab6940, Abcam, Cambridge, UK). Digital images were acquired using a Zeiss LSM 880 inverted microscope with a 60× oil-immersion objective. Fluorescence signals were detected with Cy2 (for γ-H2AX) at an excitation wavelength of 490 nm and an emission wavelength of 525 nm, and with DAPI at an excitation wavelength of 405 nm and an emission wavelength of 461 nm. All images were processed using FIJI software 1.53.

### 2.11. Statistical Analysis

Dose–response curves for estimating the cell viability were plotted by taking the logarithms of the concentrations of the compound along the X-axis and the cell viability along the Y-axis. Using GraphPad Prism Software 8.0.2, Inc., La Jolla, CA, USA), the parameter “Log (inhibitor) vs. normalized response” was chosen to estimate the IC_50_ value (concentration of a drug giving a half-maximal inhibitory response). The results are expressed as mean ± standard error (SD) or standard error of the mean (SEM). GraphPad Prism Software 8.0.2 was used for statistical analysis. The differences between two groups were analyzed by using one-way ANOVA, and the differences between groups were analyzed by using two-way ANOVA with the Dunnett and Tukey multiple comparison tests. The statistical significance is indicated as follows: * *p* < 0.05, ** *p* < 0.01, *** *p* < 0.001, and **** *p* < 0.0001.

## 3. Results

### 3.1. LAH Inhibits Colon Cancer Cell Growth 

The treatment with LAH of colon cancer cells demonstrated a marked increase in cellular inhibition with IC_50_ values of 7.184 µM and 20.35 µM for HCT116 and SW620, respectively, while remaining undetermined at 24 h in the non-tumoral CRL1790 cell line, suggesting the need for higher concentrations to achieve inhibition in non-tumor cells (Figure 1B, Appendix A). Such results suggest the potential selectivity depending on the cancer type and the potential therapeutic use of LAH for colon cancer. However, acetogenins, including LAH, have shown antiproliferative activity across various cancer types.

To understand how LAH affects cellular survival, we assessed its impact on the apoptosis in CRC and non-tumor cell lines by analyzing apoptosis-associated proteins through Western blotting. We observed an increase in the cleavage of PARP-1 in LAH-treated HCT116 and CRL-1790 cells but not in SW620 cells. The AIF levels increased in all three cell lines following the treatment, while caspase-8 showed variable expression: no change in HCT116, a decrease in SW620, and an increase in CRL-1790. The cleavage of caspase-3 (key proapoptotic protein) was observed in HCT116 and SW620 cells but was absent in the non-tumor CRL-1790 cell line (Figure 2A).

These findings highlight that LAH selectively induces apoptosis in cancer cells, as evidenced by caspase-3 cleavage, while sparing non-tumor cells. LAH is known for inhibiting mitochondrial complex I, which may initiate a cascade of events that trigger the apoptotic pathway. The disruption of mitochondrial function can lead to the release of pro-apoptotic factors and the activation of the caspase cascade, contributing to cell death. Additionally, mitochondrial dysfunction can result in the increased production of ROS, which may cause DNA damage and trigger γ-H2AX formation, an indicator of DNA double-strand breaks (DSBs).

Immunostaining for γ-H2AX clearly demonstrated that LAH treatment induces the formation of γ-H2AX foci in CRC cell lines (Figure 2B), which indicates the presence of DNA damage. This observation links mitochondrial inhibition, DNA damage, and apoptosis, suggesting that the impact of LAH on mitochondrial complex I may play a key role in initiating these downstream cellular processes, thereby enhancing its therapeutic potential against colon cancer. These findings contrast with those observed in the non-tumoral cell line, CRL1790. Further investigation is necessary to fully elucidate the intricate relationship between mitochondrial inhibition, DNA damage, and apoptosis in the mechanism of action of LAH.

### 3.2. LAH Disrupts the Mitochondrial Function and Dynamics in Colon Cancer Cells, Leading to Bioenergetic Failure

LAH not only induces DNA damage and apoptosis but also disrupts the mitochondrial function and dynamics in colon cancer cells. Mitochondria, being central regulators of programmed cell death, respond to cellular stress by modulating reactive oxygen species (ROS) levels. This mitochondrial dysfunction affects critical processes such as electron transport chain (ETC) activity, ATP production, and mitochondrial membrane potential (ΔΨ_M_ or MMP).

To investigate the specific effects of LAH on the mitochondrial function in colon cancer cells compared to non-tumorigenic cells, we assessed ΔΨ_M_ as a key indicator of mitochondrial health using tetramethylrhodamine methyl ester (TMRM), a cationic, lipophilic dye that accumulates in active mitochondria with a high membrane potential. In untreated (non-fluorescent) cells, the red histogram shows no detectable signal. However, upon incubation with TMRM, a fluorescence signal is generated, shifting the histogram to the right (blue histogram), indicating an increase in MMP.

To ensure the validity of these results, we employed FCCP (carbonyl cyanide-p-trifluoromethoxyphenylhydrazone) and oligomycin as controls. FCCP, a known mitochondrial uncoupler, disrupts the proton gradient throughout the inner mitochondrial membrane, causing a shift in the TMRM signal to the left, consistent with decreased MMP. Conversely, oligomycin, an ATP synthase inhibitor, blocks the proton flow through the F₀F₁-ATPase, causing a shift further to the right, indicating increased MMP due to proton accumulation. In the presence of LAH, a substantial increase in TMRM fluorescence intensity was observed in all three cell lines (represented by the forest green histogram) (Figure 3A,B), signifying an elevation in ΔΨ_M_. This heightened MMP likely reflects impaired oxidative phosphorylation (OXPHOS), leading to proton accumulation, increased glycolytic flux, and enhanced ROS production.

To further explore the impact of LAH on mitochondrial function, we employed the Seahorse XF Cell Mito Stress Assay to measure the key parameters of mitochondrial respiration, including basal respiration, ATP-linked respiration, proton leak, spare respiratory capacity, and maximal respiration (Figure 3C).

Our analysis revealed a significant reduction in the basal mitochondrial oxygen consumption rate (OCR) across all three cell lines treated with LAH, compared to the untreated controls (Figure 3D). This reduction suggests that LAH impairs the essential mitochondrial processes required for sustaining cellular energy production. Additionally, the OCR linked to ATP production was markedly lower in LAH-treated cells compared to both untreated and RAP-treated cells, further corroborating the impairment of mitochondrial ATP synthesis. Proton leak, a measure of protons escaping through the inner mitochondrial membrane without contributing to ATP synthesis, also increased in the LAH-treated cells, indicating mitochondrial dysfunction.

Upon the addition of FCCP, which uncouples the ETC to reveal maximal respiratory capacity, no significant increase in OCR was detected in the LAH-treated cells. This contrasts with the robust OCR increase seen in the control and RAP-treated cells, where FCCP stimulated maximal respiration. The lack of FCCP-induced OCR elevation in the LAH-treated cells signifies a depleted spare respiratory capacity, suggesting severe bioenergetic damage to mitochondria. The measurement of maximal respiration, after FCCP administration, was significantly impaired in the LAH-treated cells, a critical indicator of compromised mitochondrial function. This impairment suggests that LAH disrupts the mitochondria’s ability to meet the energy demands under stress (Figure 3D).

In conclusion, LAH significantly disrupts mitochondrial respiration at multiple levels. The observed reductions in basal respiration, ATP-linked respiration, and maximal respiration, combined with the increased proton leak, underscore LAH’s profound impact on mitochondrial integrity and bioenergetic capacity. These findings highlight the severe metabolic and bioenergetic deficits induced by LAH in colon cancer cells, providing further insight into its mechanism of action and potential as a therapeutic target.

### 3.3. LAH-Induced Disruption of the Mitochondrial Electron Transport and Energy Production in CRC Cells

To further understand the impact of LAH on mitochondrial function, we analyzed the expression of OXPHOS proteins. OXPHOS complexes are integral to mitochondrial energy production and are directly affected by changes in MMP. In HCT116 cells, LAH treatment led to a reduction in the expression of complex I and complex IV, which are critical components of the mitochondrial electron transport chain (Figure 4A,B). This reduction in OXPHOS proteins correlates with the observed decrease in MMP, as disruption in the electron transport chain can lead to decreased ΔΨ_M_ and impaired ATP production. In contrast, SW620 cells exhibited minimal changes in OXPHOS proteins and MMP, similar to non-tumor CRL-1790 cells, indicating that LAH’s effect is more pronounced in certain cancer cell types. In all the cell lines, no significant alterations were detected in complexes III and V upon LAH treatment (Appendix A).

The ATP levels, reflecting mitochondrial energy production, were significantly reduced in HCT116 cells following LAH treatment compared to SW620 and CRL-1790 cells (Figure 4C). This reduction is consistent with the observed decrease in MMP and OXPHOS protein expression, underscoring the impact of LAH on mitochondrial function and energy metabolism.

### 3.4. LAH on the Glycolytic and Mitochondrial Function in Colon Cancer Cells

Next, we evaluated the glycolytic rate. To isolate the contribution of glycolysis to proton efflux, we measured cellular OCR and subtracted it from the total proton efflux rate (PER), resulting in the glycolytic proton efflux rate (glycoPER). This value reflects the rate of protons extruded during glycolysis. Figure 5A,B show that the rapamycin (RAP)-treated and untreated cell lines exhibited low basal glycolytic rates, whereas LAH treatment led to a significant increase in the glycolytic rate compared to the controls. This enhanced glycolysis is due to LAH’s inhibition of OXPHOS, which forces cells to rely more on glycolysis for energy. The RAP-treated cells displayed similar basal proton efflux rates to the untreated controls, indicating a minimal impact on the ETC activity and glycolytic metabolism. In contrast, LAH treatment resulted in a significant rise in basal proton efflux, highlighting its pronounced effect on metabolic pathways. Figure 5C further demonstrates that LAH-induced proton efflux was approximately 50% higher than in the untreated and RAP-treated cells, indicating a substantial enhancement in glycolysis in CRC cells treated with LAH.

### 3.5. Mitochondrial Dynamics and Function Altered by LAH

One of the reasons for the altered mitochondria function can be related to changes in the mitochondrial dynamics. We examined the mitochondrial dynamics using MitoTracker staining. Mito tracker, featuring a thiol-reactive chloromethyl group for mitochondrial labeling, is a cationic lipophilic dye. Its positive charge enables penetration through the cell membrane due to lipophilicity and accumulation within the mitochondrial matrix, facilitated by the negative mitochondrial membrane potential.

Observations of the untreated cells revealed a variety of mitochondrial shapes, including spherical, rod-shaped, and filamentous arrangements, indicative of mitochondrial dynamics and the associated increased ATP production (Figure 6A).

We used HCQ and Rapamycin (RAP) as control treatments to understand the effects of LAH. HCQ, an autophagy inhibitor, affects lysosomal pH and prevents the fusion of autophagosomes and lysosomes. After treatment with 50 μM HCQ, mitochondria exhibited similar shapes to those observed in the untreated cells, suggesting that HCQ effectively inhibits mitophagy (Figure 6A). In contrast, RAP inhibits mTOR signaling and induces mitochondrial fusion, leading to elongated mitochondrial networks (hyperfusion), which protects mitochondria from degradation due to starvation effects (Figure 6A).

LAH treatment, however, resulted in significant mitochondrial fragmentation (Figure 6A). This fragmentation is associated with impaired OXPHOS, mtDNA damage, and increased ROS creation, as discussed earlier. Additionally, NaO staining revealed an increase in mitochondrial mass, suggesting that the fragmentation observed is related to alterations in mitochondrial network integrity and bioenergetic capacity, rather than a simple reduction in mitochondrial content (Figure 6B,C). This fragmentation can result from heightened mitochondrial biogenesis as a compensatory response to stress or damage, leading to a higher overall mitochondrial content. 

These findings underscore the complex effects of LAH on mitochondrial morphology and function, contrasting with the effects of other treatments and highlighting the disruption of mitochondrial network integrity and bioenergetic function induced by LAH.

### 3.6. Association between Mitochondrial Fragmentation and Mass Increase

The results indicate an increase in the mitochondrial mass in all three cell lines after 24 h of LAH treatment, likely due to induced fragmentation. This fragmentation is associated with a compensatory increase in mitochondrial content. To further investigate, Western blot analysis was performed to assess the proteins involved in the mitochondrial dynamics, including mitofusin 1 (MFN1) and dynamin-related protein 1 (DRP1), a GTPase that modules mitochondrial fission upon phosphorylation. A significant decrease in MFN1 expression was observed in SW620 and CRL-1790 cells but not in HCT116 cells (Figure 6D, Appendix A). Conversely, a significant increase in phosphorylated DRP1 was detected in LAH-treated cancer cell lines, indicating enhanced mitochondrial fission. This suggests that LAH-induced fragmentation and the associated increase in mitochondrial mass are more pronounced in cancer cells compared to non-tumor cells, reflecting a selective impact on the mitochondrial dynamics in colon cancer cells. Thus, LAH had a profound impact on mitochondrial function and dynamics, which seems to be more specific for colon cancer cell lines and does not significantly disturb non-tumor cells.

### 3.7. mTORC1 Activity and Autophagy Induction Following LAH Treatment

To elucidate the effects of LAH on mTOR complex 1 activity, we analyzed the phosphorylation status of one of its key substrates, eukaryotic translation initiation factor 4E-binding protein 1 (4E-BP1). LAH treatment significantly reduced 4E-BP1 phosphorylation in HCT116 cells and SW620 in contrast to the non-tumoral cell line, CRL1790. The inhibition effects are similar to the downregulation observed with the mTORC1-specific inhibitor Torin 1 (Figure 7A, Appendix A). Interestingly, LAH did not significantly affect 4E-BP1 phosphorylation in HCT116 cells, which suggests that changes in 4E-BP1 activity might influence the mTORC1 activity in these cells. HCQ, an autophagy inhibitor that does not act on mTORC1, showed no impact on 4E-BP1 phosphorylation. We also examined other substrates, such as p70 S6 kinase (p70S6K) and transcription factor EB (TFEB); however, p70S6K did not show any significant changes following LAH treatment, and although TFEB expression diminished in their phosphorylated form, it did not reach statistical significance (data not shown). These findings suggest that the effects of LAH on mTORC1 signaling are specific to certain substrates, further highlighting the selective nature of its action.

### 3.8. Induction of Autophagy by LAH in Colon Cancer Cells

Since LAH inhibits phosphorylated 4E-BP1, it is plausible that it selectively targets mTORC1, potentially affecting autophagic flux. To investigate this, we assessed autophagy induction using fluorescent detection. The cells were transfected with the mCherry-eGFP-LC3B plasmid and treated with Torin 1, HCQ, and LAH for 24 h. This assay visualizes autolysosomes as red dots, indicating autophagic activity.

Treatment with Torin 1 (an mTORC1 inhibitor and autophagy inducer), HCQ, and LAH resulted in the formation of mCherry-positive dots (Figure 7B). Notably, while both Torin 1 and HCQ induced mCherry-positive dots, their mechanisms differ. Torin 1 stimulates autophagy by inhibiting mTORC1, leading to increased autophagosome formation, whereas HCQ inhibits the fusion of autophagosomes with lysosomes, resulting in the accumulation of autophagic vacuoles. This distinction highlights the varied effects of the different treatments in autophagy regulation.

Further investigation into the impact of LAH on autophagy in the HCT116 and SW620 cell lines was conducted using Western blot analysis of autophagy-related proteins, including ULK1, P62, LC3I, and LC3II. ULK1 is a key kinase initiating autophagy, P62 is involved in the ubiquitinated cargo deliverance to autophagosomes, and LC3I and LC3II are crucial for autophagosomes, with LC3II serving as a marker for autophagosome formation.

The results showed a significant increase in the phospho-ULK1 levels in HCT116 cells, while a decrease was observed in the non-tumoral cell line. SW620 cells showed elevated total ULK1 levels, whereas no significant changes were noted in the HCT116 or non-tumor cells (Figure 7C, Appendix A). Additionally, LAH treatment increased the LC3II levels across all three cell lines, indicating autophagy induction; however, there were no significant changes in p62 detection (Figure 7D, Appendix A). These findings suggest that LAH induces autophagy in tumor cells, potentially through the induction of mitophagy.

## 4. Discussion

In this study, we explored the effects of a plant-derived metabolite, namely the acetogenin, LAH, on colon cancer cells, focusing on its impact on cellular energetics and mitochondrial dynamics. Our data revealed how LAH influences the mitochondrial function and metabolic pathways in CRC.

A hundred years ago, Otto Warburg discovered that tumor cells exhibit higher glycolytic activity than normal cells, suggesting mitochondrial damage and OXPHOS inhibition in tumors [22]. However, recent studies have challenged this view, showing intact mitochondrial function in most tumors and even increased OXPHOS activity in certain cancers, which supports cell proliferation, metastasis, and therapy resistance [23]. For instance, Ren et al. found that the upregulation of prohibitin 2 (PHB2) enhances mitochondrial complex I activity, boosting OXPHOS levels and promoting CRC tumorigenesis. PHB2 deletion reduces OXPHOS levels in CRC cells [24]. Similarly, in this study, we found elevated cellular OCR in CRC cells, indicating that these cells produce more ATP through upregulated mitochondrial OXPHOS rather than glycolysis.

LAH impaired mitochondrial function, characterized by the reduced detection of OXPHOS complexes [14,25]. This results in ATP depletion [26] and altered mitochondrial dynamics. This impairment significantly impacts cellular metabolism. Our findings demonstrate that LAH treatment leads to a notable decrease in the expression of OXPHOS complexes, critical for mitochondrial energy production. This reduction in OXPHOS proteins correlates with a decrease in ΔΨ_M_ and a significant drop in ATP levels, confirming the disruption of mitochondrial function. Additionally, the observed changes in mitochondrial dynamics and metabolic shifts underscore LAH’s profound effect on cellular energy production and the overall metabolic state.

OXPHOS dysfunction reduces mitochondrial ATP generation, leading to increased glycolysis [27]. Our study supports this, showing decreased ATP levels following LAH treatment (Figure 3 and Figure 4). Additionally, LAH significantly impaired mitochondrial respiration and decreased the reserved respiratory capacity, leading to severe mitochondrial dysfunction. These findings align with previous studies on various natural compounds that disrupt mitochondrial respiration and induce compensatory glycolysis in cancer cells [28,29,30].

Understanding the relationship between energy metabolism and mitochondrial dynamics is crucial in cancer biology [31]. Mitochondria maintain a continuous balance between fusion and fission [32]. Mitochondrial fusion hastens the exchange of materials in the mitochondria to compensate for mitochondrial function [31], and mitochondrial fission eliminates impaired mitochondria by way of mitophagy [33]. The induction of mitochondrial fragmentation and an increase in the mitochondrial mass suggest a complex interplay between these dynamics and the metabolic responses triggered by LAH. Mitochondrial fragmentation, often associated with dysfunctional mitochondria [34] and increased ROS production, may contribute to cellular stress and apoptosis [33].

The mitochondrial fusion at the outer mitochondrial membrane is mediated by MFN1 and mitofusin 2 (MFN2), whereas the fusion at the inner mitochondrial membrane is mediated by optic atrophy protein 1 (OPA1) [35]. Mitochondrial fission is mediated by DRP1, FIS1, and MFF (31). Under LAH treatment, we did not observe a reduction in MFN1 expression. This discrepancy may be attributed to the potential compensatory role of MFN2, which, to some extent, can substitute for MFN1 [36] in mitochondria lacking MFN1 due to their structural similarity [37].

Polyphilin I, a natural product obtained from Paris polyphylla rhizomes, induces the mitochondrial translocation of DRP1, leading to mitochondrial fission, the release of cytochrome c from the mitochondria to the cytosol, and eventual apoptosis [38]. We observed an increase in mitochondrial fission, as evidenced by the elevated levels of p-Drp1 protein in CRC cells exposed to LAH, but not in the non-tumor line. These findings accord with MitoTracker staining, where we observed that the LAH treatments induced mitochondrial fragmentation in the CRC cell lines. Mitochondrial fusion has been observed in response to treatments with RAP and HCQ [39].

This mitochondrial fragmentation is attributed to cellular dysfunction [40], OXPHOS impairment [41], mtDNA damage [42], and ROS production [43]. Moreover, mitochondrial fragmentation facilitates the autophagic elimination of damaged mitochondria and is closely associated with mitochondrial fission [44]. In contrast, our findings reveal that untreated cells typically exhibit a large tubular mitochondrial array, extending throughout the cytosol and closely interacting with the nucleus and endoplasmic reticulum [45]. The highly dynamic mitochondrial networks oscillate between a fragmented state and a tubular continuum, reflecting a balance between mitochondrial fusion and fission, thereby promoting increased ATP production [46]. Under the conditions of metabolic stress or nutrient depletion, the rapid and transient elongation of the mitochondrial network occurs, termed “stress-induced mitochondrial hyperfusion” [47]. Mitochondrial hyperfusion has been observed in response to treatments such as RAP and CQ [39], while LAH result in fragmented mitochondria, suggesting potential mitochondrial fission.

Additionally, the dynamic alterations in mitochondrial energy have been linked to dysregulation in MMP [48]. ΔΨ_M_ serves as a crucial indicator of mitochondrial function, and ΔΨ_M_ originates from proton pumping across the inner mitochondrial membrane concomitant with electron transport [49]. This process drives various essential mitochondrial functions, such as ATP synthesis, calcium accumulation, and the maintenance of ion gradients [48]. Previous studies reported a gradual decrease in ΔΨ_M_ in HepG2 cells upon exposure to LAH [14]. Our results present contrasting data. We observed an increase in ΔΨ_M_ in the CRC cells treated with LAH across all three cell lines. This increase in ΔΨ_M_ may contribute to diminish ATP production and promote ROS generation. Changes in ΔΨ_M_ can influence the alterations in energy metabolism. 

In addition to the observed alterations in mitochondrial dynamics and energy metabolism, our study sheds light on the involvement of key cellular processes such as mTOR signaling, apoptosis, and autophagy in the response to treatment with LAH. The dysregulation of mTOR signaling has been associated with multiple types of cancer, including CRC, where it plays a central role in promoting cell growth and proliferation [50,51,52,53]. In our study, we observed a decrease in phosphorylated 4E-BP1 indicating mTOR inhibition.

mTOR inhibition has been shown to promote events such as apoptosis and autophagy. The acetogenin analogue, AA005, induces autophagy through the activation of AMPK, thereby blocking the mTOR pathway and leading to the increased expression of LC3 [16]. These findings align closely with our own data, wherein we observe the up-regulation of LC3II, suggesting an augmentation in the origination of the ATG12-ATG5 conjugate, a pivotal step in autophagosome biogenesis.

Under prolonged stress conditions, autophagy may also contribute to programmed cell death. ACGs induce cell cycle arrest in the G1/S phase [54,55,56] through the deregulation of various pathways such as MAPK [57] or Notch [18]. These pathways can increase the ROS levels [56], alter energy metabolism [17], and induce apoptosis, either in a caspase-dependent or caspase-independent manner [15,19]. We found a significant induction of programmed cell death in CRC cells following LAH treatment. LAH triggers apoptosis in CRC cells through multiple mechanisms. Firstly, we observed an increase in DNA DSBs, as shown by the upregulation of phosphorylated γ-H2AX, a well-established marker of DNA damage and a key component of the DNA damage response (DDR) [58]. γ-H2AX formation occurs at sites of DNA DSBs, where it serves as a recruitment platform for various DDR proteins involved in repair and signaling processes [59]. Previously, it was shown that the natural compound, resveratrol, which inhibits complex I of the ETC [60], sensitized radioresistant prostate cancer cells by increasing ATM phosphorylation and its target protein γ-H2AX, resulting in cell cycle arrest and subsequently cell death [61].

DDR activation initiates a cascade of events aimed at repairing the damaged DNA or triggering apoptotic cell death if repair is not feasible. The increased γ-H2AX signal observed in our study reflects the cellular response to the genotoxic stress inflicted by LAH, highlighting the effectiveness of LAH in inducing DNA damage and activating apoptotic pathways in CRC cells. Furthermore, our study revealed a cascade of events following the induction of DNA DSBs, notably the activation of caspase cascades. The observed increase in caspase-3 activation corroborates the initiation of the apoptotic signaling pathways triggered by LAH-induced DNA damage. This coordinated response underscores the potency of LAH in inducing apoptotic cell death in CRC cells through the activation of both the DNA damage response and apoptotic pathways. These results are similar to those reported by our group [10], who demonstrated that LAH induces cell death via apoptosis in a murine cancer colon model, correlating these results with decreased migration and a reduction in tumor volume (10). Similarly, LAH exhibits greater efficacy in decreasing tumor volume compared to doxorubicin [9].

As is well known, working with natural products presents certain limitations. The principal concerns include the bioavailability and access to the primary source, which in this case was the *A. macroprophyllata* tree. Our research group addressed these issues by enlisting an expert botanist to ensure the collection of sufficient fruit for laherradurin extraction.

The chemical complexity of acetogenins provides them with a diverse range of bioactivity, making them promising candidates for the development of anticancer agents. However, the isolation and characterization of these metabolites pose additional challenges. Improving the extraction method and synthesizing these compounds will be crucial to meet the demand for chemotherapeutic agents in cancer treatment and to ensure the safety of these compounds. Furthermore, the toxicity of acetogenins represents another area that requires further investigation; additional preclinical tests must be conducted to eventually progress to clinical trials in humans.

The discovery of drugs from natural products has proven to be a successful and reliable approach for proposing new structures to combat various illnesses. In this article, we demonstrate for the first time the versatility of laherradurin in inducing cancer cell death in colon cancer cell lines, revealing autophagy induction and mitochondrial dysfunction as mechanisms of action.

LAH exerts a range of effects on CRC cells, including significant alterations in mitochondrial function, metabolic reprogramming, and the induction of apoptosis. Our study demonstrates that LAH disrupts mitochondrial function by inhibiting the key components of the electron transport chain, leading to an increased reliance on glycolysis and subsequent changes in cellular energy metabolism. This metabolic shift not only affects tumor cell viability but also enhances the autophagic processes, which may further influence LAH’s antitumor efficacy.

Future research should focus on elucidating how LAH interacts with specific metabolic pathways and signaling networks within CRC cells. Investigating the molecular details of LAH-induced mitochondrial damage, its impact on cellular energetics, and the precise role of autophagy and mitophagy in LAH’s therapeutic effects will be essential. Furthermore, exploring LAH’s effects in diverse in vivo models and clinical trials will be crucial for assessing its potential as a novel anticancer agent. Understanding these aspects could significantly advance the development of targeted therapies for CRC, addressing the metabolic vulnerabilities and potentially improving patient outcomes. By translating these findings into clinical practice, we may uncover new strategies to combat CRC and enhance the effectiveness of the existing treatments.

## 5. Conclusions

Our study provides compelling evidence for the therapeutic potential of LAH in CRC treatment. Our findings reveal that LAH exerts its anticancer effects through the alteration of mitochondrial function via the inhibition of OXPHOS complexes, accompanied by increased glycolysis and mitochondrial fragmentation. The disruption of mitochondrial dynamics and bioenergetics contributes to cellular stress and the induction of apoptosis, as evidenced by DNA DSBs and the activation of apoptotic pathways. In addition, LAH downregulates mTORC1 signaling and induces autophagy. In summary, LAH is emerging as a promising therapeutic agent against CRC. Further clarification of the precise molecular mechanisms underlying its effects, along with their validation in preclinical and clinical settings, will be crucial for their translation into clinical practice. Such clarification deserves future studies.

## Figures and Tables

**Figure 1 cells-13-01649-f001:**
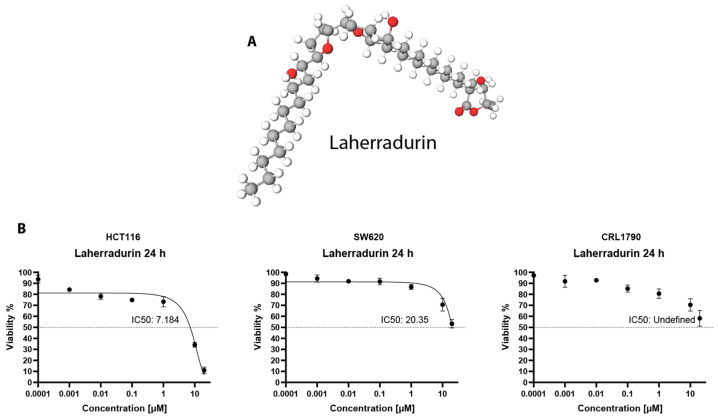
Dose-dependent effects of LAH on cellular inhibition. (**A**) Chemical 3D structure of LAH. Created by using MolView 3D software v2.4. Source: PubChem. (**B**) Dose–response curves showing IC_50_ values of CRC and non-tumoral cells treated with LAH during 24 h exposure, assessed via SRB. Curves represent triplicate biological repeats and are displayed as mean ± SEM (*n* = 3).

**Figure 2 cells-13-01649-f002:**
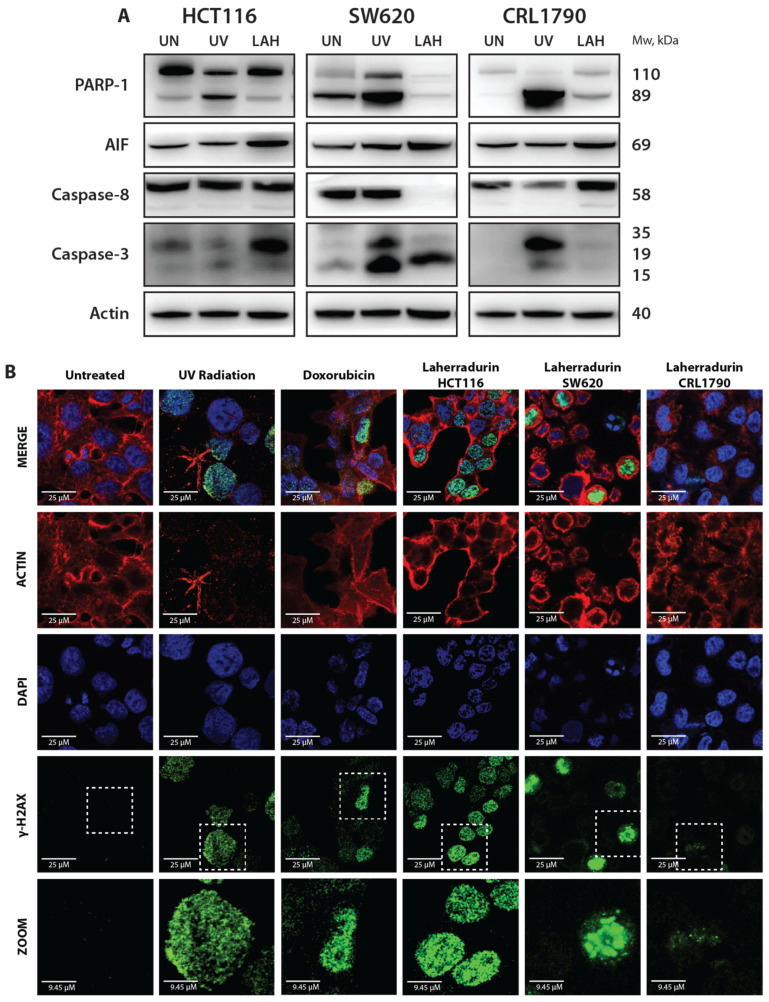
LAH induces apoptosis and DNA damage. (**A**) Representative images of protein expression levels of PARP-1, AIF, Caspase-8, and Caspase-3. GAPDH was used as loading control. Replicates were conducted three times in total. (**B**) H2AX foci detection in CRC and non-tumor cells. Fluorescence images taken with Zeiss LSM 880 confocal microscope at 63×. γ-H2AX, DAPI, ACTIN, and ZOOM with treatments: Untreated, UV radiation, Doxorubicin, LAH at 24 h exposure. The photos are representative of at least three independent experiments. Scale bar: 20 μm.

**Figure 3 cells-13-01649-f003:**
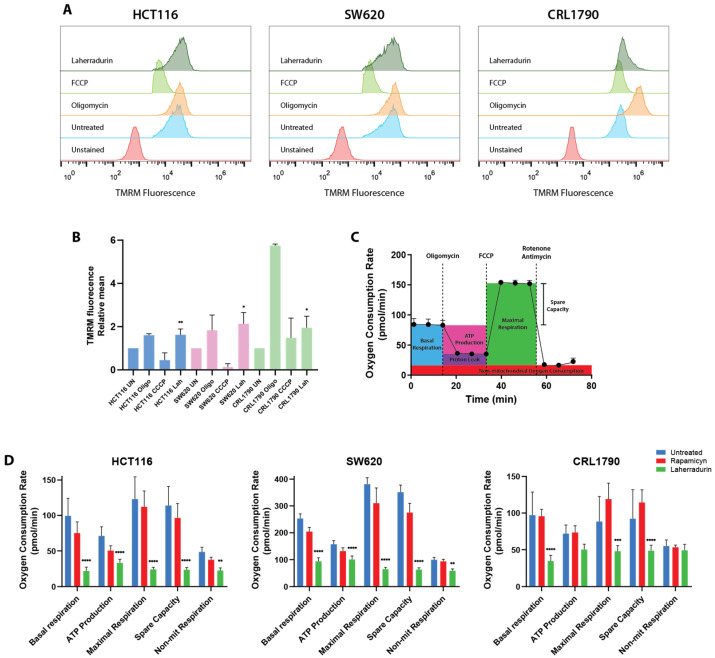
ΔΨ_M_ Disruption and Bioenergetic Alterations. Cells were stained for TMRM (indicator ΔΨ_M_) and analyzed in flow cytometry. (**A**) Representative cytograms depicting the relative intensity of TMRM in CRC cells and non-tumoral cells treated with LAH for 24 h. (**B**) Bar graph shows relative TMRM intensity of CRC cells treated with LAH at 24 h. Results represent at least three independent biological experiments. Statistical significance relative to the control was determined by one-sample *t*-test. ** *p* < 0.05, * *p* < 0.1 (**C**) Schematic representation illustrating oxygen consumption measured through extracellular flux analysis, highlighting the calculable parameters. (**D**) Bioenergetic profile comprising five respiration parameters. OCR following sequential addition of various modulators of mitochondrial function (Oligomycin, FCCP, and Rotenone/Antimycin A) was determined using a Seahorse XF96 analyzer. Cells were treated with LAH individually for 24 h. The results are compiled from at least three independent biological experiments, each involving independent cell seeding in eight wells of Seahorse microplates. Data are presented as mean ± SE. Statistical significance relative to control is indicated as follows: **** *p* < 0.001, *** *p* < 0.01, ** *p* < 0.05, as determined by two-way ANOVA with Dunett’s multiple comparisons test.

**Figure 4 cells-13-01649-f004:**
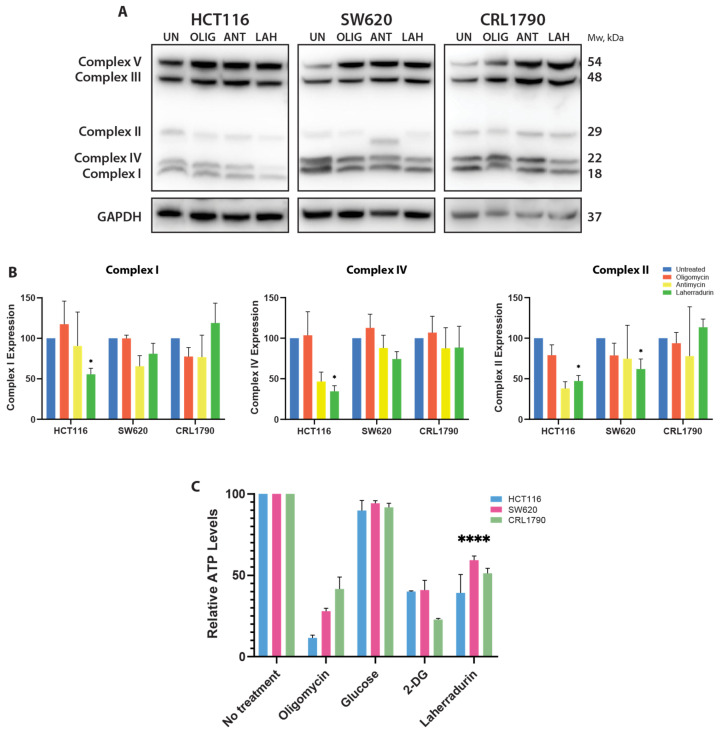
Expression of OXPHOS complexes in Mitochondria from CRC and Non-Tumor Cells. (**A**) A cocktail of antibodies targeting proteins representing the five mitochondrial OXPHOS complexes was utilized to assess the expression of mitochondrial proteins in CRC cell lines, under the following experimental conditions: Untreated (UN), Oligomycin, Antimycin A (ANT), and LAH. GAPDH was used as a loading control. (**B**) The antibody cocktail included the following targets: complex I subunit NDUFB8, complex II subunit SDHB, complex III subunit UQCR2, complex IV subunit MTCO1, and complex V subunit ATP5A. The WB was technically replicated three times in total, demonstrating a representative blot. Values are expressed as mean ± SEM normalized relative to their respective control and then presented as fold change relative to treatment: * *p* < 0.1. A two-way ANOVA test with Dunett’s multiple comparisons test was applied. (**C**) Relative ATP levels of cells were measured at 24 h of treatment. Treatment groups included: No treatment, Oligomycin, Glucose, 2-DG, and LAH. Results are presented as the mean ± SE of three independent experiments. Statistical significance was determined by a two-way ANOVA with Tukey’s multiple comparisons test (**** *p* < 0.001).

**Figure 5 cells-13-01649-f005:**
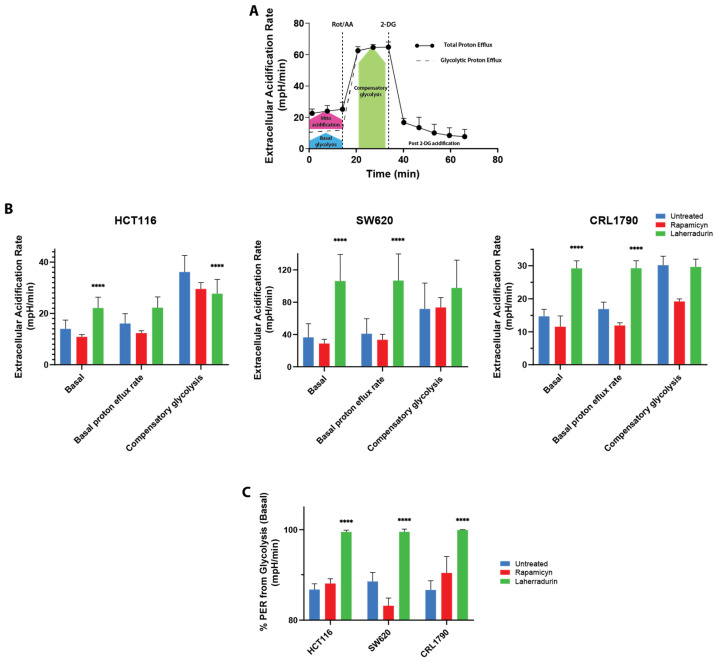
Analysis of glycolytic flux in CRC and non-tumor cells. (**A**) Schematic of proton efflux rate from extracellular flux analysis showing the calculable parameters. ECAR after sequential addition of different modulators of glycolytic function (Rotenone/Antimycin A and 2-DG) was determined using a Seahorse XF96 analyzer. (**B**) Bioenergetic profile of glycolytic parameters. Basal: basal respiration. Basal proton efflux rate: (glycoPER) glycolytic basal rate. Compensatory glycolysis: the ability of the cell to compensate for energy production through glycolysis following mitochondrial ATP production blockage. (**C**) % PER of glycolysis indicates the contribution of the glycolytic pathway to total extracellular acidification. Results represent at least three independent biological experiments, each of which consisted of independent seeding of cells in eight wells of Seahorse microplates. Data are presented as mean ± SE. **** *p* < 0.001, relative to control by two-way ANOVA with Dunett’s multiple comparisons test.

**Figure 6 cells-13-01649-f006:**
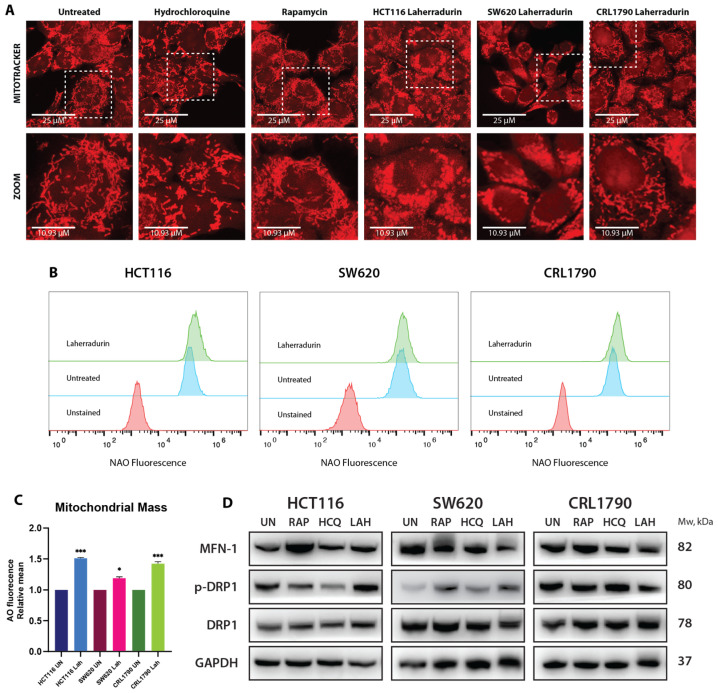
Effects of natural compounds on mitochondrial dynamics and function in CRC and non-tumor cells. (**A**) Fluorescence images of mitochondrial morphology stained with MitoTracker Red in CRC and non-tumor cells. Images were captured using LEICA confocal microscope at 63× magnification. Treatments include: Untreated, RAP, CQ, and LAH with 24 h exposure. Representative images from at least three independent experiments. Scale bar: 25 μm; zoom: 10.93 μm. (**B**) Representative cytograms depicting the relative intensity of NaO in CRC cells and non-tumoral cells treated with LAH for 24 h. (**C**) Bar graph illustrating the relative NaO intensity of CRC cells treated with LAH for 24 h. Cells were stained with NaO and analyzed via flow cytometry. Data represent at least three independent biological experiments, presented as mean ± SE. Statistical significance was assessed using one-sample *t*-test, with *** *p* < 0.01, and * *p* < 0.1, relative to the control. (**D**) Representative images and quantification of protein expression levels of MFN1, p-DRP1, and DRP1, normalized to GAPDH as a loading control.

**Figure 7 cells-13-01649-f007:**
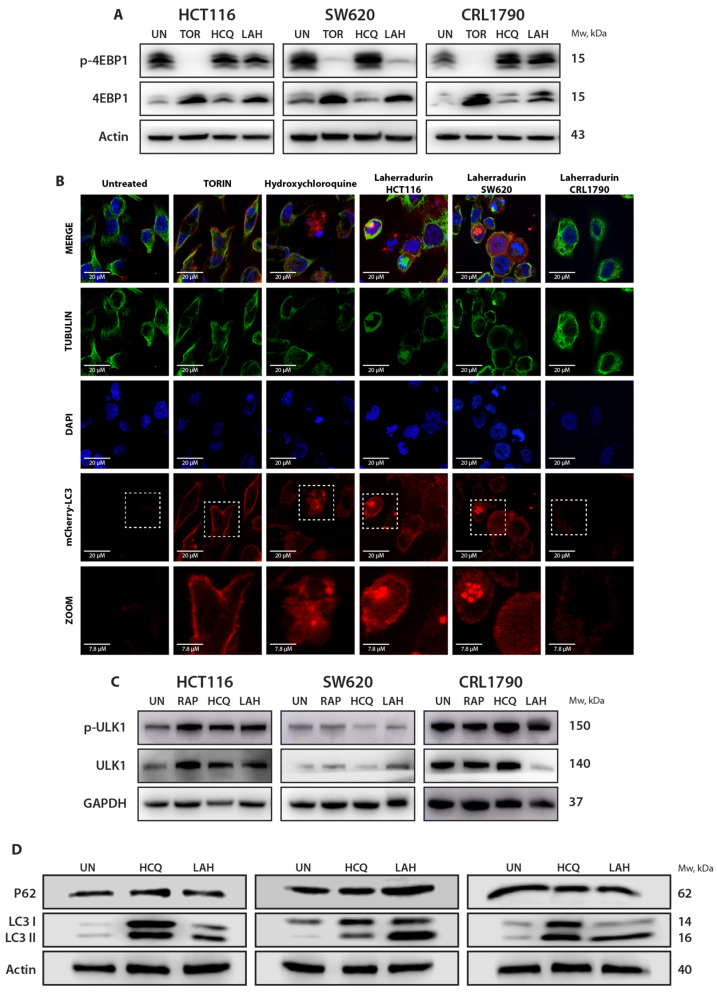
LAH modulates mTOR pathway and induces autophagy. (**A**) Representative images and graphical quantification of protein expression levels of p-4E-BP1, and 4E-BP1. (**B**) Autophagosome detection stained with mCherry in CRC and non-tumor cells. Fluorescence images taken with Zeiss LSM 880 confocal microscope at 63×. mCherry-LC3, Tubulin, DAPI, and ZOOM with treatments: Untreated, RAP, CQ, LAH at 24 h exposure. The photos are representative of at least three independent experiments. Scale bar: 20 μm. (**C**,**D**) Representative images of protein expression levels of p-ULK, ULK, P62, LC3-I, and LC3-II. GAPDH and actin were used as loading controls. Replicates were conducted three times in total.

## Data Availability

All data is available on request to Nadia Jacobo-Herrera (nadia.jacobo@gmail.com, nadia.jacoboh@incmnsz.mx) and Carlos Perez-Plasencia (carlos.pplas@gmail.com).

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
