# Peer review of "Laherradurin Inhibits Colorectal Cancer Cell Growth by Induction of Mitochondrial Dysfunction and Autophagy Induction"

_cells, 2024, doi:10.3390/cells13191649_

Round 1

Reviewer 1 Report

Comments and Suggestions for Authors

Delgado-Waldo and co-authors submitted the manuscript "Laherradurin inhibits colorectal cancer cell growth by inducing mitochondrial dysfunction and inhibiting autophagy". The manuscript is a regular article.

Laherradurin (LAH) is an acetogenin from the Annonaceae family, with antitumor activity already demonstrated in various tumor cell lines and mouse models in previous publications. However, the authors explored in detail the mechanism of action of LAH by evaluating the inhibition of the electron transport chain and oxidative phosphorylation (OXPHOS) complex, thus triggering mitochondrial damage, alteration of the mitochondrial membrane potential, and apoptosis in two CRC cell lines. Moreover, the authors demonstrated that treatment with LAH induces mitochondrial fission and increased mitophagy. Interestingly, the effects of LAH in cancer cells are minimal in the non-tumor line, suggesting a potential use of LAH in colorectal cancer therapy.

To achieve the results, the authors used several experimental approaches, including flow cytometry, western blotting, mitochondrial respiration, immunocytochemical analysis with confocal microscopy. The description of the related methodologies is clear and detailed. The results are reported clearly and intelligibly and are convincing in supporting the hypotheses advanced by the authors. Therefore, I have no important criticism to raise with the authors.

Minor:

1) The manuscript requires only minor adjustments in English in a few sentences.

Author Response

Reviewer 1:

Dear Reviewer,

We sincerely appreciate your time and attention in reviewing our manuscript. We have meticulously checked the English to enhance the quality of the article.

Reviewer 2 Report

Comments and Suggestions for Authors

This study introduces a promising idea about inhibiting colorectal cancer cell growth by inducing mitochondrial dysfunction and autophagy inhibition by Laherradurin. After minor revision, this can be accepted.

In Fig. 1, with the dose-response curves about the middle, the IC50 was calculated as 20.35 uM, and the maximum percentage of cell viability didn't exceed 50. 

- Did the authors confirm the reliability of the IC50 calculation by counting the cells stained by Trypan Blue to ensure that 50% of the cells died? 

However, the manuscript contains good experimental design and presentation. 

Comments on the Quality of English Language

No comments regarding Language

Author Response

Dear Reviewer,

Thank you for your valuable comments and observations.

Q1: In Fig. 1, the dose-response curves indicate that the IC50 was calculated as 20.35 µM, while the maximum percentage of cell viability did not exceed 50%. 

Answer: Only in the SW620 cell line, the concentrations used in the cytotoxicity assay did not allow the experimental determination of an ic50 (with the highest concentration corresponding to 20 micromolar we obtained 53.2932% cell viability, data represented by the last dot in the graph). However, employing the statistical analysis of dose response estimation curve with Prism 8 GraphPad software, we obtained an IC50 value of 20.35 micro molar as shown in the attached image .

Q2: Did the authors confirm the reliability of the IC50 calculation by counting the cells stained with Trypan Blue to ensure that 50% of the cells died? 

Answer: We did not employ the Trypan Blue technique to determine cell viability; instead, we relied on the sulforhodamine B assay as an adequate indicator of viability in conjunction with laherradurin treatment. Furthermore, we conducted sulforhodamine B experiments in triplicate, which were then re-evaluated in a double-blind manner by a second researcher.

Reviewer 3 Report

Comments and Suggestions for Authors

The manuscript presents an original and thorough investigation on the effects of the natural compound Laherradurin (LAH) on mitochondrial function and autophagy in colorectal cancer (CRC) cell lines.

The Introduction is well focused and structured and based on previous research of the team on LAH biological activities. The present goal is focused on examining the antitumor effects of LAH in CRC cells.

The Materials and Methods section is presented in detail and the methodology is easily reproducible. A variety of informative and adequate to the aim of the study assays have been used.

Cytotoxicity tests, Western blotting and Immunoflorescence were applied together with several techniques assessing mitochondrial respiration, membrane potential and morphology. This complex methodological approach provides the basis for the reliable results and conclusions obtained.

The Results are illustrated by 7 highly informative composite figures combining bioenergetics, morphology, Western blot and immunofluorescence. The results are presented concisely. They show changes in mitochondrial dynamics, inhibited mitochondrial complex I activity, ATP depletion and increased glycolysis. Mitochondria are fragmented and autophagy is activated. As caspase-3 is shown to be cleaved, apoptosis is selectively triggered resulting in reduced CRC cell vitality.

The Discussion section is analytical and comprehensive. A short paragraph on the limitations of the study could be recommended.

The Conclusions are based on the results obtained. The authors claim ‘ In addition, LAH down-regulates the mTORC1 signaling and induce autophagy’ while in the title they state ‘ Laherradurin inhibits colorectal cancer cell growth by induction of mitochondrial dysfunction and autophagy inhibition’.   How would they explain this contradiction as results show that LAH induces autophagy in CRC cells?

The English language and the overall style of the manuscript need no linguistic revision. The references are relevant and are correctly stated.

In conclusion, the manuscript offers a detailed study on the effects of LAH on mitochondrial function and the mechanism of cell death in CRC cell lines.

The manuscript is suitable for publication after minor clarifications. 

Author Response

Dear Reviewer,

Thank you for your insightful comments and observations, which will help us enhance the article. In response, we have added a brief paragraph to the discussion addressing the limitations of the study as suggested.

Q1. Limitations of the Study: 

As is well-known, working with natural products presents certain limitations. The principal concerns include bioavailability and access to the primary source, which in this case was the A. macroprophyllata tree. Our research group addressed these issues by enlisting an expert botanist to ensure the collection of sufficient fruit for laherradurin extraction.

The chemical complexity of acetogenins provides them with a diverse range of bioactivity, making them promising candidates for the development of anticancer agents. However, the isolation and characterization of these metabolites stand additional challenges. Improving the extraction method and synthesizing these compounds will be crucial to meet the demand for chemotherapeutic agents in cancer treatment and to ensure the safety of these compounds. Furthermore, the toxicity of acetogenins represents another area that requires further investigation; additional preclinical tests must be conducted to eventually progress to clinical trials in humans.

The discovery of drugs from natural products has proven to be a successful and reliable approach for proposing new structures to combat various illnesses. In this article, we demonstrate for the first time the versatility of laherradurin in inducing cancer cell death in colon cancer cell lines, revealing autophagy induction and mitochondrial dysfunction as mechanisms of action.

Q2: The conclusions are based on the results obtained. The authors state, “In addition, LAH down-regulates the mTORC1 signaling and induces autophagy,” while the title claims, “Laherradurin inhibits colorectal cancer cell growth by induction of mitochondrial dysfunction and autophagy inhibition.” How would you explain this contradiction, given that results show that LAH induces autophagy in CRC cells? 

Answer: We appreciate your observation regarding the contradiction between the title and conclusions, which was indeed an oversight on our part. The title should read: "Laherradurin inhibits colorectal cancer cell growth by induction of mitochondrial dysfunction and autophagy induction." We will revise the title to accurately reflect our findings.

Thank you once again for your constructi